# Management of Appendix Neuroendocrine Neoplasms: Insights on the Current Guidelines

**DOI:** 10.3390/cancers15010295

**Published:** 2022-12-31

**Authors:** Amr Mohamed, Sulin Wu, Mohamed Hamid, Amit Mahipal, Sakti Cjakrabarti, David Bajor, J. Eva Selfridge, Sylvia L. Asa

**Affiliations:** 1Division of Hematology and Medical Oncology, UH Seidman Cancer Center, 11100 Euclid Avenue, Cleveland, OH 44106, USA; 2Department of Internal Medicine, UH Seidman Cancer Center, 11100 Euclid Avenue, Cleveland, OH 44106, USA; 3Department of Medical Genetics, Center for Human Genetics, Case Western Reserve University, Cleveland, OH 44106, USA; 4Department of Stem Cell Biology and Regenerative Medicine, City of Hope Beckman Research Institute, Duarte, CA 91010, USA; 5Department of Pathology, UH Seidman Cancer Center, Case Comprehensive Cancer Center, Case Western Reserve University, Cleveland, OH 44106, USA

**Keywords:** management, appendix, neuroendocrine, neoplasms

## Abstract

Appendiceal neuroendocrine neoplasms (ANENs) usually present as incidental findings at the time of appendectomy for acute appendicitis. They are rare, accounting for only 0.5–1% of intestinal neoplasms; they are found in 0.3–0.9% of all appendectomy specimens. They are usually sporadic tumors. There are several histological types including well-differentiated neuroendocrine tumors (NETs), poorly differentiated neuroendocrine carcinomas (NECs), and mixed neuroendocrine-non-neuroendocrine neoplasms (MiNENs). Histologic differentiation and the grade of well-differentiated NETs correlate with clinical behavior and prognosis. Management varies based on differentiation, aggressiveness, and metastatic potential. There is debate about the optimal surgical management for localized appendiceal NETs that are impacted by many factors including the tumor size, the extent of mesoappendiceal spread, lymphovascular invasion and perineural involvement. In addition, the data to guide therapy in metastatic disease are limited due to the paucity of these tumors. Here, we review the current advances in the management of ANENs within the context of a multidisciplinary approach to these tumors.

## 1. Introduction

The most common appendiceal tumors are neuroendocrine neoplasms (ANENs), followed by mucinous neoplasms and adenocarcinomas [1]. ANENs account for 25–60% of primary malignancies in the appendix [2,3,4,5,6,7]. Most cases present as incidental findings; they are found in 0.16–2.3% of appendectomies performed for acute appendicitis [8,9,10]. The most common age at diagnosis is at the end of the second decade of life with an increased incidence in females [11,12,13,14,15,16]. According to the current WHO histological classification, ANENs include well-differentiated NETs, poorly differentiated NECs (large cell and small cell types), and mixed neuroendocrine-non-neuroendocrine neoplasms (MiNENs) [17,18,19]. The majority of ANENs are well-differentiated NETs (70–75%) that are further subdivided into different grades (G1, G2 and G3) according to their proliferative rate [19]. Grade 1 tumors are usually indolent tumors with relatively long median survival durations compared to high-grade tumors, which are more aggressive [20,21]. Poorly differentiated NECs, which resemble small-cell or large-cell neuroendocrine carcinomas of the lung, have aggressive behavior, and usually present with metastatic disease at diagnosis [18,19,22,23,24]. MiNENs are the rarest tumor type. ANENs are heterogeneous tumors, and their management depends on various factors including histological differentiation, disease stage, hormone production, somatostatin receptor expression, tumor burden, and hepatic versus extrahepatic disease. This review article focuses on the clinical presentation, staging workup, and current treatment guidelines of different types of NENs arising in the appendix.

## 2. Epidemiology

The incidence of NENs including ANENs has been increasing in the past decade, partly attributed to a better understanding of the pathophysiologic presentation, improved classification system, and availability of advanced imaging modalities [25]. The annual incidence of ANENs is reported to be around 3–9 cases per 1000 appendectomies [11,26,27]. The age at diagnosis ranges from 20 to 50 years in adults, and most patients are in their 40s, which is younger than the average age for other gastroenteropancreatic neuroendocrine neoplasms (GEP-NENs) and other primary malignant appendiceal neoplasms [3,5,28]. There is a relatively higher incidence in women; this has been attributed to the greater frequency of incidental appendectomies in women who undergo pelvic surgery [16,20,29].

ANENs are most often identified at the distal tip of the appendix; the next most common location is the body and they are only rarely present at the base of the structure [2], where they can cause complications due to obstruction [2,30,31,32]. Based on the SEER database of information collected between 1973 and 2004, 60% of ANENs are localized diseases at diagnosis, while 28% and 12% of cases have regional and distant metastasis, respectively [15]. About 50% of patients affected by poorly differentiated or undifferentiated tumors presented with synchronous distant metastasis at diagnosis [15].

## 3. Classification of Appendiceal Neuroendocrine Neoplasms

The 2019 World Health Organization (WHO) Classification of Tumors of the Digestive System categorizes neuroendocrine neoplasms into two broad subgroups: well-differentiated neuroendocrine tumors (NETs) and poorly differentiated neuroendocrine carcinomas (NECs) [19]. Well-differentiated NETs, formerly often called “carcinoid tumors”, are further subdivided into grades 1, 2, and 3 according to the proliferative rate determined by the Ki67 labeling index and/or the mitotic index (MI). Well-differentiated grade 1 tumors are considered relatively indolent, grade 3 tumors are more aggressive but still better than poorly differentiated NECs, which are high-grade aggressive carcinomas (small cell or large cell carcinomas) with poor outcome (median survival rate less than 2 years) [33]. Approximately 70 percent of NENs within the appendix are well-differentiated NETs [20,21]. A third category, mixed neuroendocrine-non-neuroendocrine neoplasms (MiNENs), includes tumors that are composed of a mixed population of neuroendocrine neoplasm and adenocarcinoma; currently, it is suggested that there must be a minimum of 30% of the tumor mass composed of the smaller component however this remains controversial [19]. Both the neuroendocrine and non-neuroendocrine components may have varying differentiation and grade. The median age at diagnosis of MiNENs is approximately 58 years. These tumors comprise up to 6.9% of ANENs and have a 5-year survival rate of 56.3% (95% CI, 42.1–68.4) which is lower than well-differentiated NETs but better than NECs [20,34]. Despite the fact that a large meta-analysis showed that the most common primary site of MiNEN is the appendix (60%), the data on these tumors is limited [35,36]. The tumor formerly known as “goblet cell carcinoid” is now recognized to be an adenocarcinoma with dominant mucin-secreting cells and a minor component of neuroendocrine cells; these are no longer considered among ANENs [37].

Localized well-differentiated NETs have median overall survival greater than 20 years, the best prognosis among all GEP-NENs; the overall prognosis is highly variable according to tumor morphology, size and stage. Small, low-grade ANENs localized to the appendix have a better prognosis than large ANENs with high-grade morphology, and tumors with extra-appendiceal invasion or metastasis [30]. Among well-differentiated NETs, grade 1 is the most common and many are smaller than 1 cm, which accounts for good survival rate [38]. The NCDB-based study has demonstrated that ANETs have a 5-year survival rate of 86.3% (95% confidence interval [CI], 81.4–89.9) and low rates of regional and distant metastases [20]. An association between tumor size and both 5-year survival rate and metastatic potential was reported in this study. The 5-year survival rate decreases as the lesion size increases 89.9%, 70.6% and 58.2% for tumor size ≤2 cm, 2–4 cm, and >4 cm, respectively [20]. In-addition, both regional and distant metastases are less commonly seen with tumors smaller than 2 cm. This information is the basis for the TNM-staging system for appendiceal NETs.

While the majority of appendiceal NETs are EC cell tumors producing serotonin, there are also less common L cell tumors that produce glucagon-like peptides, pancreatic polypeptides and Peptide YY [18,39,40]. L cell tumors may have unusual morphology, with predominant tubular or clear cell architecture. Accordingly, the product of the tumor can be important to be known for surveillance. However, it remains to be seen if the cell type affects prognosis as it does in rectal NETs [41].

## 4. Clinical Presentation

The majority of appendiceal NETs are asymptomatic and are found incidentally at the time of appendectomy. The reason for an appendectomy is usually acute appendicitis or chronic non-specific lower right quadrant abdominal pain. Because most of these tumors are located in the distal appendix, they are unlikely to cause obstruction; only 10% are located at the base of the appendix and can be implicated as the cause of obstruction and appendicitis [2,30,31,32]. Tumors larger than 2 cm and at the base of the appendix are associated with a higher incidence of appendicitis due to obstruction as well as higher rates of nodal and distant metastases [30,31,32]. Carcinoid features may be present in patients with tumors that have metastasized to the liver but they are very uncommon (<1%) [15,42]. As observed in midgut GEP-NET cases, symptomatic serotonin-producing NET is associated with mesenteric fibrosis, which could cause bowel obstruction. However, the evidence of relevant mechanisms underlying the rare obstruction identified in ANEN remains unavailable.

Patients with high-grade poorly differentiated NECs and MiNENs usually present with metastatic disease at the time of diagnosis [11,30,43]. They are hormonally inactive tumors with no associated symptoms of hormone hypersecretion. The most common signs and symptoms are non-specific and related to the size and location of the tumor; they include pain, nausea, fatigue, anorexia, weight loss, abnormal liver function test results, and bowel obstruction if associated with bowel metastasis [2,30,31,32].

## 5. Diagnostic Assessment

Most ANENs are found incidentally, therefore, the diagnostic work-up should focus on accurate histopathological assessment as well as biochemical and imaging evaluations to detect early recurrence and/or metastasis.

## 6. Circulating Biomarkers

Chromogranin A (CgA) is a protein that is stored and released with peptides and amines in NETs [44,45,46]. Several reports have shown that elevated levels are associated with recurrence, and levels twice the normal were associated with poor outcomes and shorter survival in NETs [47,48,49,50]. Some guidelines advocate surveillance with serum levels of CgA; this biomarker is often non-specific as it can be elevated in other medical conditions (renal and hepatic insufficiency, uncontrolled hypertension), and with certain medications (proton pump inhibitors and steroids) [51,52]. However, these comorbidities are less common in the population of younger patients with ANETs. Nevertheless, they must be considered when using surveillance with CgA.

Measurement of the serotonin metabolite 5-hydroxyindoleacetic acid (5-HIAA) in plasma or a 24 h urine collection can be considered, particularly in patients with liver metastasis and carcinoid features such as diarrhea, flushing, and wheezes. It is not indicated in tumors that do not produce serotonin and patients without carcinoid features [53,54,55].

## 7. Staging and Imaging

The role of staging anatomical imaging is mainly to identify nodal and distant metastasis. Given that tumor size is a strong factor in determining the likelihood of regional and distant metastases, the follow-up recommendations depend on tumor size, the status of resection margins and lymph nodes and mesoappendiceal invasion. Most patients with small well-differentiated appendiceal NETs <2 cm with negative margins and no deep mesoappendiceal invasion (<3 mm) will have a low risk of recurrence and most guidelines recommend no surveillance for them [26,56,57]. For patients with an appendiceal NET >2 cm, incomplete resection, and positive nodes or margins, the current guidelines recommend workup with contrast-enhanced, triple-phase computed tomography (CT) scan or magnetic resonance imaging (MRI). However, as discussed below, a more sensitive technique for this assessment is somatostatin receptor-based diagnostic positron emission tomography [PET] scanning, and it is important for patients with metastatic NETs. According to the North American Neuroendocrine Tumor Society (NANETS) and the European Neuroendocrine Tumor Society (ENETS) guidelines, somatostatin receptor (SSTR)-based imaging is not necessary for patients with localized disease but this is a position that might warrant reconsideration. Its role in advanced stages is to assess receptor status and other metastatic lesions that were not identified with anatomical scans [58,59]. Octreotide SPECT/CT (single-photon emission computed tomography) is less sensitive and SSTR-PET (somatostatin receptor-based PET) tracers (either gallium Ga-68 or copper CU-64 dotatate) are preferred. SSTR-PET is considered the gold standard in diagnosis and surveillance, and it is currently used for patients with symptoms suggestive of carcinoid syndrome and/or metastatic diseases.

Measurable lesions are considered SSTR-positive if the uptake is greater than that of the liver. SSTR-PET scan is usually combined with anatomical imaging (triple phase CT or MRI) to identify both SSTR-positive and -negative lesions.

In patients with high-grade, poorly differentiated NEC, ^18^F-FDG-PET combined with triple-phase CT or MRI is preferred, given that these de-differentiated carcinomas usually lack SSTR expression [60,61,62,63]. Several studies of functional and anatomical imaging for neuroendocrine tumors have shown that ^18^F-FDG PET/CT has sensitivity and specificity of 61.9% and 100%, while 68Ga-DOTATATE PET/CT has sensitivity and specificity of 100–81% and 90–80%, respectively [64,65,66,67]. The false positive rate of 68Ga-DOTATATE PET/CT varies between 0 to 38%, likely due to tracer accumulation in lymph nodes containing SSTR2+ macrophages, inflammatory cardiovascular tissues and other regions affected by chronic inflammatory conditions [65,68,69].

## 8. Management

Much of the data on the management of metastatic ANETs is based on large trials that include all GEP-NETs. There have been only a very few retrospective single-institution reports of ANETs [70,71,72].

### 8.1. Treatment of Localized Disease

The majority of patients are diagnosed after a simple appendectomy, and further surgical intervention will depend on tumor size and the presence of high-risk features (deep mesoappendiceal invasion >3 mm, positive/unclear margins, positive lymphovascular invasion, and a higher proliferative rate). The incidence of metastasis is associated with increases in the size of ANET lesions; 2 cm lesions are found to have nodal involvement in up to 33% of cases and distant metastasis in 12% [73]. Accordingly, the consensus-based guidelines from the NANETS and the ENETS recommend a simple appendectomy for any lesion smaller than 1 cm and for those tumors between 1.0 and 1.9 cm that lack high-risk features [74,75,76]. The optimal surgical approach for localized well-differentiated ANETs between 1–2 cm with high-risk features is controversial [77,78]. The NCCN guidelines consider appendectomy alone to be adequate for all tumors <2 cm, even with high-risk features [2,27]. Other reports demonstrate a higher potential for nodal and metastatic spread with small tumors ≥1.5 cm [57,79]. Analysis of the National Cancer Institute (NCI) Surveillance, Epidemiology, and End Results (SEER) database between 1988 and 2003 identified significantly higher rates of lymph node metastases in patients with appendiceal NETs larger than 2 cm compared with those with either between 1.0–1.9 cm or less than 1 cm (86% vs. 47% and 15%, respectively) [26]. The results were confirmed in another analysis of the SEER database between 1988 and 2013 that included 573 patients with well-differentiated ANETs. The results reported a 64% probability of nodal metastases in patients with tumors >2 cm vs. 31% in those with tumors measuring 1.1–2 cm [80].

Despite this higher risk of nodal metastasis in lesions measuring 1.0–1.9 cm compared to lesions smaller than 1 cm, a series from the National Cancer Database (NCDB) did not show a survival benefit with the addition of right hemicolectomy for patients with lesions smaller than 2 cm, even in those with high-risk features (5-year survival 88.7 versus 87.4%) [68,74,81] [Table 1]. Therefore, simple appendectomy is considered sufficient therapy for lesions smaller than 1 cm and right hemicolectomy only is recommended for lesions larger than 2 cm in most guidelines. For lesions between 1.0 and 2.0 cm, a multidisciplinary approach is essential to discuss several factors including high-risk features, lesion site (base vs. apex), patient age, comorbidities, and the likelihood of surgical complications. If a right hemicolectomy is planned, a full colonoscopy should be performed to rule out synchronous colorectal cancers and a full inspection of the bowel intraoperatively is suggested to identify other lesions [12,68,74,75,82,83] [Table 2]. Other retrospective studies and single institution experiences showed that following NANETs and ENETS guidelines may lead to overtreatment of patients RHC and suggested detailed counseling regarding the risk of over- and undertreatment needs in these patients [70,71,72,84].

According to most of the available guidelines, patients with small tumors (<1 cm) with no aggressive features who were treated with appendectomy with clear margins (R0) do not need active surveillance and no surveillance imaging is required [82,83]. However, patients with symptoms of hormone hypersecretion warrant further evaluation for disease recurrence. Surveillance is also not required for patients with ANENs that measure between 1 and 2 cm if they received the right hemicolectomy and no residual disease was identified on histological examination [Figure 1]. NANETs guidelines recommend against surveillance imaging for all small well-differentiated ANENs (<2 cm); in contrast, other guidelines, including ENETS, recommend follow-up for patients with tumors measuring 1–2 cm that were incompletely resected, had lymph node involvement, exhibited lymphovascular invasion and/or were higher grade tumors (G2 or G3) [9,16,29,57]. The recommendation for these situations is surveillance with biochemical markers in selected patients and anatomical imaging with either a CT scan or MRI. Regarding biochemical markers, yearly chromogranin A (CgA) has been recommended with caution given its non-specificity. 24 h urinary or plasma 5HIAA can be used to monitor patients with serotonin-producing tumors and has been used for patients with clinical symptoms of carcinoid syndrome; its value in the surveillance of asymptomatic patients remains to be proven. There are no studies of the addition of other biomarkers such as glucagon, pancreatic polypeptide or PYY that are expressed by L cell tumors.

A series of studies reported by Moertel et al. have had a great influence on the current treatment recommendations by reporting limited to no recurrence in ANET patients with tumor size ≥2 cm after appendectomy [2,85]. The larger tumor size of well-differentiated ANET is known to associate with a higher rate of lymph node involvement, 11.6–16.7% vs. 29.9–56.8% vs. 40.6% for tumor size <1 cm, 1–2 cm, and >2 cm, respectively [25,86]. The rationale of RHC in the treatment of ANET >2 cm or with high-risk features is to remove regional lymph nodes associated with the elevated risk of recurrence or distant metastasis to improve overall survival. However, Groth et al. has reported no significant differences in overall survival between appendectomy and RHC for ANET >2 cm [87]. Furthermore, unlike pancreatic and other GEP-NETs, various ANET studies have shown no differences in the rate of regional lymph node positivity, metastatic disease, and overall survival between different tumor sizes (>2 cm, 1–2 cm, or <1 cm). Worse survival was observed in patients with distant metastatic disease at diagnosis while regional lymph node involvement does not impact overall survival in ANET patients [77,78]. A recent meta-analysis has shown that 10-year disease-specific survival is not significantly different between patients with and without lymph node involvement, 95.6% and 99.2%, respectively (OR: 0.2, 95% CI: 0.02–2.4) [88]. ENETS and NCCN have recommended that simple appendectomy can be sufficient even for a tumor size great than 2 cm or any tumor size without the high-risk features mentioned above.

SSTR-based imaging scans (PET-Ga68 or CU-64) are not routinely recommended for the localized disease after complete resection. A reasonable follow-up strategy for selected patients involves a careful history and physical examination with biochemical markers when appropriate and anatomical imaging 6–12 months after surgery and yearly afterward for up to 10 years, given the risk of late recurrence [7,10,74,75,76,83,89,90] [Figure 1].

### 8.2. Treatment of Metastatic Disease

The first step in the management of patients with metastatic well-differentiated ANENs is to determine factors that may impact therapeutic strategies such as tumor grade, site of metastatic disease (hepatic vs. extra-hepatic), tumor burden, and the status of somatostatin expression [91,92]. This includes anatomical imaging as well as SSTR-PET scans for patients with well-differentiated NETs or ^18^F-FDG-PET scans for those with NEC, MiNEN or high-grade NET. There are several therapeutic options that can be considered, depending on the site and extent of disease; these include surgery, liver-directed therapy for localized liver disease, medical therapy with somatostatin, targeted agents or cytotoxic chemotherapy, and radiation therapy using radiolabeled somatostatin or external beam radiation. Most of the data from prospective trials are for patients with metastatic gastroenteropancreatic neuroendocrine tumors (GEP-NETs) with only a small number of ANENs in these trials.

### 8.3. Surgery

The presence of metastatic disease in NETs does not preclude surgical debulking of hepatic metastases. For patients with low and intermediate-grade tumors who are symptomatic and have mainly liver metastasis, surgical resection may palliate symptoms and improve long-term survival. The available data supporting surgical cytoreduction are exclusively retrospective, the level of evidence supporting this approach is limited and most data are for midgut and pancreatic NETs [93,94,95,96]. The degree of liver cytoreduction remains an area of debate, and most historical data adopt ≥90% threshold of cytoreduction to be associated with improved outcomes [97,98,99,100]. In contrast, other studies reported that ≥90% cytoreduction was not associated with the improved median OS when compared with at least a 70% threshold [99,101,102,103,104]. However, these are retrospective studies with potential selection bias; the current literature supports the hypothesis that debulking of liver metastases is associated with symptomatic relief leading to improved quality of life and improved survival. Therefore, surgical cytoreduction for patients with metastatic ANETs should be considered in a multidisciplinary discussion to appropriately select patients who may benefit from this approach, especially for those with grade 1 and 2 well-differentiated tumors [Figure 2]. Other sites of metastases such as peritoneal metastases are likely more common in mucinous appendiceal epithelial neoplasms as compared with ANETs. Based on data from other appendiceal neoplasms and colon cancer, debulking may improve the overall prognosis and survival of patients with exclusively peritoneal metastases but no current data are available specifically for ANETs. Due to the limited evidence of the overall course and outcome of surgical intervention in this subgroup of patients, there is no agreement or specific guideline established for the treatment of peritoneal metastasis of ANETs.

### 8.4. Liver-Directed Therapy

The regional lymph nodes and liver are the dominant sites of metastases for ANETs, and liver-directed therapies are appropriate for patients whose tumors are predominantly metastatic to the liver and surgical resection is not feasible. Data are mainly retrospective and prospective trials are extremely limited [92,96,105,106,107]. Options for liver-directed therapies include bland hepatic artery embolization, chemoembolization with intra-arterial cytotoxic agents (doxorubicin or cisplatin), and radioembolization using ^90^Y embedded in either a resin or glass microsphere. Other ablative techniques include radiofrequency ablation (RFA), microwave ablation and cryoablation therapy. Current retrospective literature and single institution experiences demonstrate ORRS and symptomatic relief in approximately 40–50% of patients, but most of these data are for GEP-NETs with only a very small number of ANENs included [42,108].

There are no randomized prospective trials comparing the various liver-directed therapy techniques, therefore the choice of technique depends on various factors including tumor size, location, grade, patient comorbidities and, most importantly, institutional experience with the various embolization modalities. Generally, ablation techniques are reserved for small metastases (less than 5 cm in diameter) compared to embolization methods, which are recommended for larger, more extensive and multilobar metastases.

Toxicities associated with these techniques vary. The common side effects of chemoembolization and bland embolization include abdominal pain, nausea, pain, fever and elevated serum transaminases due to the induction of ischemic hepatitis [108,109,110]. ^90^Y radioembolization does not induce ischemic hepatitis so pain and other side effects are lower than with other embolization techniques, but it can be associated with radiation enteritis and delayed liver fibrosis [111]. Although surgical cytoreduction is the only potentially curative therapy for ANETs with exclusively liver metastasis, these liver-directed therapies may offer effective alternatives to alleviate symptoms and potentially impact the outcome in patients with unresectable tumors.

### 8.5. Somatostatin Analog Therapy

Somatostatin is a 14-amino acid peptide that inhibits the secretion of other hormones such as serotonin, insulin, glucagon, gastrin and VIP and also has potent antiproliferative actions [112,113]. Due to the short half-life of native somatostatin, synthetic agents with longer half-life have been developed for therapeutic applications. Long-acting somatostatin analogues (SSAs) are peptide hormones that bind to somatostatin receptors (SSTR, mainly subtypes 2 & 5) to inhibit the secretion of other hormones and alleviate hormonal symptoms such as diarrhea and flushing [113,114,115]. In addition, SSAs have also been shown to inhibit tumor growth in randomized phase III trials (PROMID and CLAIRNET) and therefore are considered the main foundation for treating metastatic well-differentiated NETs [116,117,118].

The PROMID phase III study showed that octreotide long-acting repeatable (LAR) 30 mg deep intramuscular every 4 weeks improves time to progression (TTP) compared to placebo in patients with metastatic midgut NETs (14.3 months versus 6 months, HR 0.34; *p* = 0.000072) [117]. The CLARINET phase III study demonstrated similar effects with lanreotide depot 120 mg deep subcutaneously every 4 weeks. Results have shown that lanreotide depot 120 mg every 4 weeks significantly prolonged median progression-free survival (PFS) compared to placebo (not reached versus 18 months, HR 0.47; *p* = 0.001) [117,119]. The PROMID study included mainly midgut NETs but the CLAIRNET trial included midgut, pancreas, hindgut and tumors of unknown primary site. There was no specific percentage of ANETs included in either trial. Generally, SSAs are well tolerated with few side effects that have been reported including fat malabsorption, diarrhea, increase risk of gallstones, hyperglycemia, and mild bradycardia [112,120]. The presence of somatostatin receptors determined by PET gallium Ga-68 DOTATATE or Cupper-64 [Ga-68 DOTATATE] is predictive of response to SSAs [121,122]. However, in some cases, these diagnostic imaging modalities may be negative even if the tumors express SSTRs, and the use of SSAs in these cases is controversial. Whether evidence of SSTR expression is required before starting SSAs is an area of debate that need further investigation.

### 8.6. Radiolabeled Somatostatin Analog Therapy

Radiolabeled somatostatin analogue therapy (also known as peptide receptor radionuclide therapy; PRRT), is a novel form of targeted radionuclide therapy. It involves the use of a radio-labelled somatostatin analogue to deliver a targeted cytotoxic amount of radiation by emitting radioactive particles to SSTR-expressing disseminated tumors. Recently, the FDA approved the use of Lutetium-177 (177Lu)-DOTA-TATE (LUTATHERA ^®^) for the treatment of metastatic somatostatin receptor-positive (SSTR) GEP-NETs). Data from the phase III NETTER-1 trial in patients with metastatic mid-gut NETs who progressed on first-line SSA therapy showed a 79% reduction in risk of progression or death (median PFS not reached versus 8.4 months, HR 0.21; *p* < 0.00001) compared to high dose octreotide [123]. The objective response rate was only 18% but was significantly higher compared to the control group (18% vs. 3%, *p* <.001). Only one patient had an ANET as the primary tumor site in that study. The side effects of PRRT include mild nausea and vomiting, fatigue, lymphopenia, and less than 2% risk of myelodysplastic syndrome (MDS) and acute leukemia. Based on the results of the NETTER-1 study, the current guidelines recommend PRRT for patients with metastatic well-differentiated GEP-NETs including ANETs who progress on first-line SSA therapy. These patients require positive SSTR-based imaging (PET-Ga68 or CU64) given that expression of SSTRs is predictive of PRRT benefit.

### 8.7. Targeted Therapies (mTORi and Anti-Angiogenesis)

Multiple targeted therapies including mTOR, tyrosine kinase and angiogenesis inhibitors have been approved for treating advanced NETs [124]. The mTOR inhibitor Everolimus has been shown to decrease NET cell growth and therefore was studied in patients with metastatic GEP-NETs [125,126]. Several phase III trials have demonstrated that Everolimus 10 mg significantly improves PFS in patients with well-differentiated low and intermediate-grade NETs [127,128,129]. However, RADIANT 2 trial did not show a promising clinical benefit of Everolimus versus placebo in functional midgut NETs (HR 0.77, *p* = 0.026), RADIANT 4 trial met its primary endpoint with the improvement om PFS in non-functional low and intermediate-grade gastrointestinal NETs (11 months versus 3.9 months, HR 0.48, *p* < 0.00001). Based on these results, Everolimus was FDA-approved for use in several metastatic GEP-NETs including ANETS. Comparing the results of RADIANT 2 and RADIANT 4, Everolimus had only marginal benefit in patients with functional midgut NETs including ANETs. Only 1% of the patients in RADIANT trials had ANET as the primary tumor site. The most commonly reported side effects of Everolimus are stomatitis with oral ulcers, diarrhea, pneumonitis, hyperglycemia and myelosuppression [130,131]. Whether decreased dosing of Everolimus (5 mg vs. 10 mg) will result in the same clinical benefit with fewer side effects is yet to be investigated.

Because NETs are highly vascular tumors and express vascular endothelial growth factor (VEGF), multi-tyrosine kinase receptor inhibitors including those that target VEGF, platelet-derived growth factor receptors (PDGFRs), and Fibroblast growth factor receptors (FGFRs), have been studied extensively in metastatic GEP-NETs [132,133,134]. Sunitinib, which targets VEGF receptors 1, 2, and 3, is the main approved tyrosine kinase inhibitor (TKI) for metastatic pancreatic NETs. No data are available for mid-gut NETs. Recently, other multi-TKIs have been investigated and have shown promising results for extra-pancreatic NETs.

Surufatinib is a novel, oral multi-kinase inhibitor that selectively inhibits the VEGFRs and FGFRs which both inhibit angiogenesis. SANET-ep, is a placebo-controlled phase III study that investigated the role of surufatinib in advanced extrapancreatic NETs [135,136]. The study performed in China enrolled 198 patients on surufatinib 300 mg daily (n = 129) or placebo (n = 69). The primary tumor was mainly gastrointestinal but only 1% were ANETs. The trial met the primary end point in which surufatinib was associated with improved PFS compared to placebo (9.2 vs. 3.8 months, HR 0·33; 95% CI 0.22–0.50; *p* < 0·0001) [137]. The most common side effects included hypertension, proteinuria, and less commonly headache, diarrhea, and myelosuppression. The results suggest that surufatinib might be a new treatment option for advanced metastatic well-differentiated NETs pending multiregional randomized clinical trials for FDA approval. Other multi-TKIs investigated in phase II trials for advanced NETs include lenvatinib, pazobanib, and cabozatinib; they have also shown promising results but the number of ANETs included in these trials is unknown [138,139,140]. Although anti-angiogenesis and mTOR inhibitors have potential clinical benefits in GEP-NETs including ANETs, less than 10% of patients will have a radiological response. Therefore, a better understanding of the disease biology with new predictive biomarkers may lead to improvement of our current therapeutic strategies for GEP-NET subtypes.

### 8.8. Cytotoxic Chemotherapy

The role of systemic chemotherapy in metastatic well differentiated NETs is debatable and restricted to patients with high tumor burden and high-grade NETs. The most common used chemotherapy in well-differentiated NETs is oral capecitabine plus the oral alkylating agent temozolomide (CAPTEM) and oxaliplatin based regimens. This regimen was shown to be effective in both cell lines and early phase I as well as recent phase II trial (ECOG 2211) mainly in patients with pancreatic NETs [141,142]. The current data for CAPTEM in midgut NETs are mainly retrospective and most have shown suboptimal responses compared to pancreatic neuroendocrine tumors (pNETs). Only one prospective study of CAPTEM has shown a response rate of 41% in patients with “carcinoids” gulati [143], however the primary tumor sites of the 28 patients in this study were unclear. Based on current data, there is no strong evidence to establish the benefit of CAPTEM in patients with ANETs, therefore, it should be only considered in a clinical trial setting or for selective ANET patients with high-grade tumors and high tumor burden who have exhausted other options of therapy.

Oxaliplatin-based combinations (FOLFOX and CAPOX) have shown anti-tumor activity in patients with advanced GI NETs [135,144]. The current data are based on small phase II trials and retrospective analyses with a small total number of treated patients. Data from combined analysis of two-phase II trials included 76 patients with metastatic “carcinoids” who were treated with either FOLFOX or CAPOX plus bevacizumab and demonstrated minimal clinical benefit. An objective response rate of 13.6% and median PFS of 19.3 months in FOLFOX plus bevacizumab and 18% with a median PFS of 16.7 months in CAPOX plus bevacizumab [136]. There was no clear description of how many patients had ANETs in this trial. Given limited data for oxaliplatin-based regimens in midgut NETs include ANETs, it should be considered only for patients with high-grade tumors and high tumor burden who do not have other options for treatment [Figure 3].

**Extra-pulmonary poorly differentiated neuroendocrine carcinomas (NECs)** are extremely rare and the most common primary tumor locations are the pancreas, large bowel, and stomach [145]. Data regarding the prevalence of poorly differentiated appendiceal neuroendocrine carcinoma (ANEC) are scant and unreliable. Most patients with gastrointestinal NECs have metastatic disease at presentation and their main treatment is etoposide plus platinum (either cisplatin or carboplatin) which is similar to the treatment of small cell lung cancer [146,147,148]. The benefit of platinum etoposide in gastrointestinal NEC has been shown mainly in retrospective and early phase II studies. The Nordic consortium was a large retrospective study of 252 patients with advanced GEP-NEC [149]. The results demonstrated a response rate of 31%, 4 months PFS and 11 months median survival compared to those who had only supportive care. The most common site of the primary tumor was the stomach, pancreas, and colon, and 38% of patients had either “other GI” or an unknown primary site with no specific designation of ANECs. Another retrospective series included 123 GEP-NEC patients who received a platinum plus etoposide regimen for first-line treatment. The study showed an objective response rate (ORR) of 50% with a median PFS of 6.2 months and overall survival of 11.6 months [150]. It is not clear whether ANEC was part of the 20% “other GI and unknown primary” tumors included in this study. Overall, a platinum-etoposide regimen induces a response rate between 30% and 50% with a short duration (less than 9 months), poor overall survival of less than 2 years and a significant toxicity profile [151,152,153].

Recently, a phase III study evaluated Irinotecan-based regimens (irinotecan plus cisplatin doublet, IP) versus platinum etoposide (EP) in advanced NECs of the digestive system [154]. The study enrolled 170 patients with gastrointestinal and hepatobiliary NEC to either IP or EP. There were no significant differences in either PFS (5.6 vs. 5.1, HR 1.060, 95% CI, 0.777–1.445) or RR 54.5% vs. 52.5% in EP and IP, respectively. The study showed no superiority of one regimen over the other and both can be considered standard first-line therapy for metastatic gastroenteropancreatic neuroendocrine carcinomas (GEP-NEC.) There was no information about how many of the gastrointestinal NECs were ANECs.

Patients with gastrointestinal NEC who progress on platinum etoposide will have limited therapeutic options with poor outcome. There are limited data for second-line therapy and mostly derived by case series and small trials with no standard regimen has been established. Some of the available options include regimens that have been used in other GI malignancies such as Oxaliplatin- or irnotecan-based regimens (FOLFOX or FOLFIRI) [150,155,156]. Recently, several phase II trials (DART and CA209) have evaluated the benefit of dual anti-CTLA-4 and anti-PD-1 blockade in refractory extra-pulmonary NEC patients [145,153,157,158]. The results demonstrated significant clinical activity with a response rate range between 24–44%. The DART trial included only one patient with an appendiceal primary tumor, but the CA209 trial did not include patients with ANECs [Figure 4].

***MiNEN*** represents an extremely rare diagnosis, but a common primary site location is the appendix. The biological behavior is usually driven by the neuroendocrine component, which is most often a poorly differentiated NEC. Therefore, MiNENs are treated similarly to pure high-grade NEC. Surgery can be considered for potentially curable early-stage disease; palliation for symptom relief is the approach for advanced-stage diseases. In one retrospective study, the median recurrence-free survival for surgery in localized disease was 12.5 months [159]. There is very limited data about the benefit of neoadjuvant and/or adjuvant therapy and minimal guidance for the choice of regimen. Two of the most used regimens in clinical practice are either EP or oxaliplatin-based (FOLFOX), but neither is supported by randomized prospective evidence. Retrospective analyses and single-institution experience have shown that multimodal treatment (including chemotherapy and/or radiotherapy) vs. surgery alone was associated with a significantly prolonged OS (75.0 vs. 18.9 months, *p* = 0.0045) [159]. These results demonstrated the benefit of multimodal treatment over surgery alone in localized MiNEN.

For metastatic disease, the data are also limited. Treatment is similar to pure NEC with EP as the main regimen gave the aggressive nature of the disease [33,148,160]. The available results are exclusively retrospective and are similar to pure NEC with ORR between 30% and 40%, and median PFS of approximately 4–5 months [33,161]. There is a paucity of data regarding second-line therapy and most common regimens are irinotecan-based regimens (FOLFIRI, or Irinotecan plus cisplatin) with limited benefit. There are no similar data for dual ICPIs available for appendiceal MiNEN [Figure 5].

## 9. Conclusions

Appendiceal NENs (ANENs) are a heterogeneous group of neoplasms with biological behaviors that range from indolent to highly aggressive. Due to limited data, every case requires a multi-disciplinary discussion to include clinical and pathological characteristics that will impact their management. Current surgical strategies for localized well-differentiated ANETs may vary from simple appendicectomy to right hemicolectomy depending on tumor size and patterns of invasion. Because the diagnosis is usually established at the time of histological examination of an appendectomy specimen, it is important to identify the subgroup of patients who will require further therapy. Recently, there have been significant advances in the management of NETs and novel therapeutic options are available for patients with metastatic disease; cytotoxic chemotherapy should be used only for patients with high-grade NETs and high tumor burden who have no other therapeutic options. Although poorly differentiated ANECs are extremely rare, their management involves platinum etoposide chemotherapy. Very few trials have included patients with ANENs and the data are scant. Therefore, it is crucial to evaluate many factors including surgical respectability, even in the metastatic setting, status of somatostatin avidity, histological grade, and tumor burden to select the appropriate personalized treatment for every patient through multidisciplinary discussion by NET experts.

## Figures and Tables

**Figure 1 cancers-15-00295-f001:**
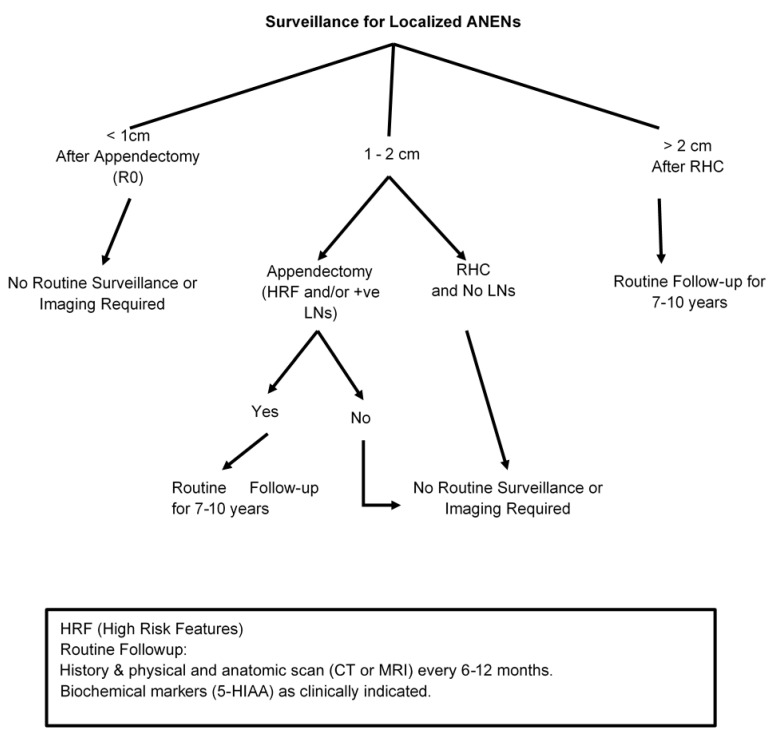
Post-treatment Follow-up After Surgery for Appendiceal Neuroendocrine Neoplasms (ANENs).

**Figure 2 cancers-15-00295-f002:**
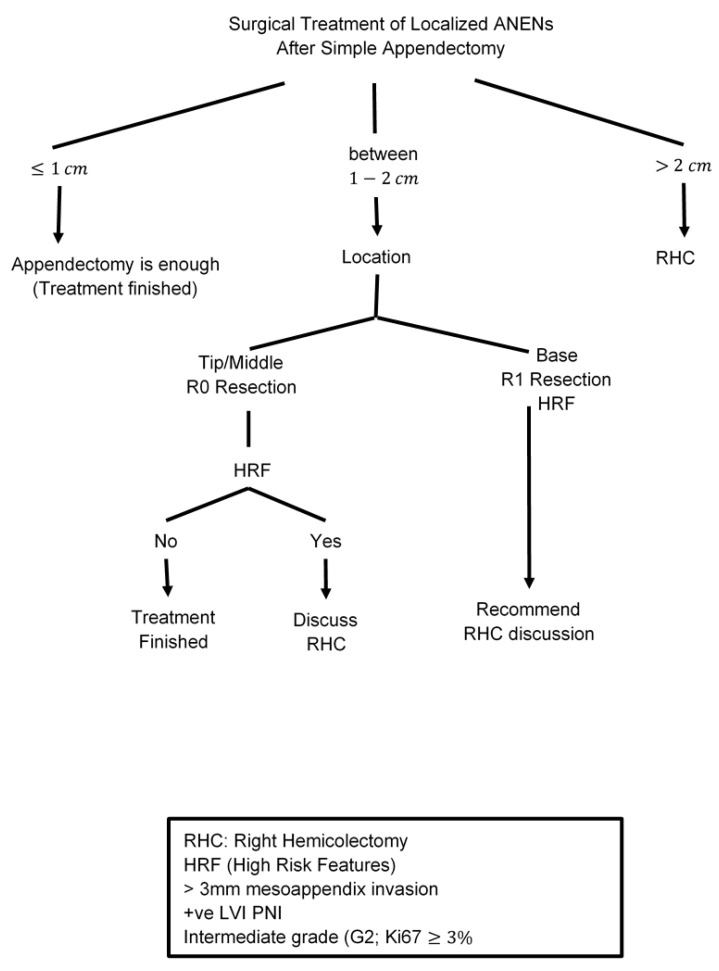
Surgical Treatment of Localized ANENs after Simple Appendectomy.

**Figure 3 cancers-15-00295-f003:**
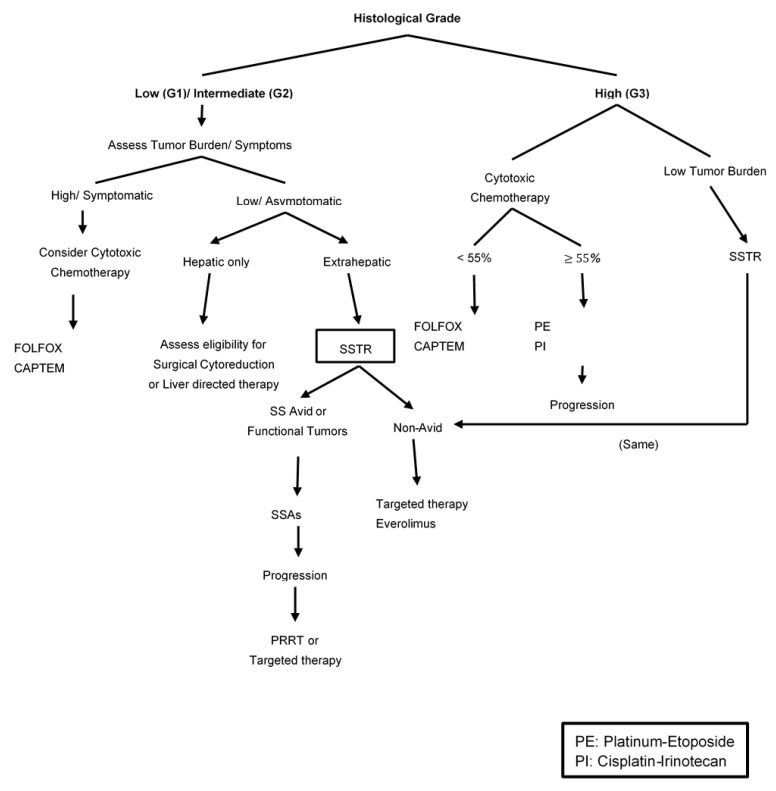
Therapeutic Strategy for Metastatic Well-Differentiated ANENs based on histological grades.

**Figure 4 cancers-15-00295-f004:**
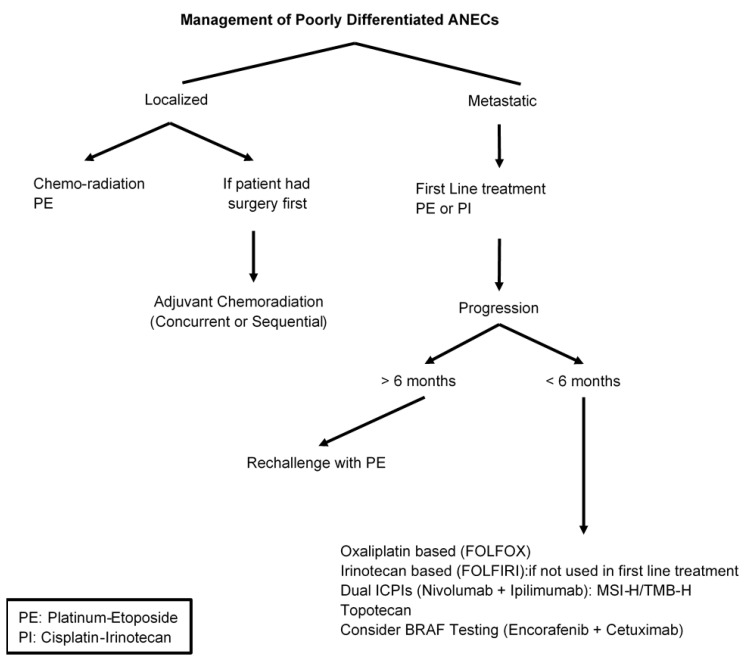
Management of Poorly Differentiated ANEC.

**Figure 5 cancers-15-00295-f005:**
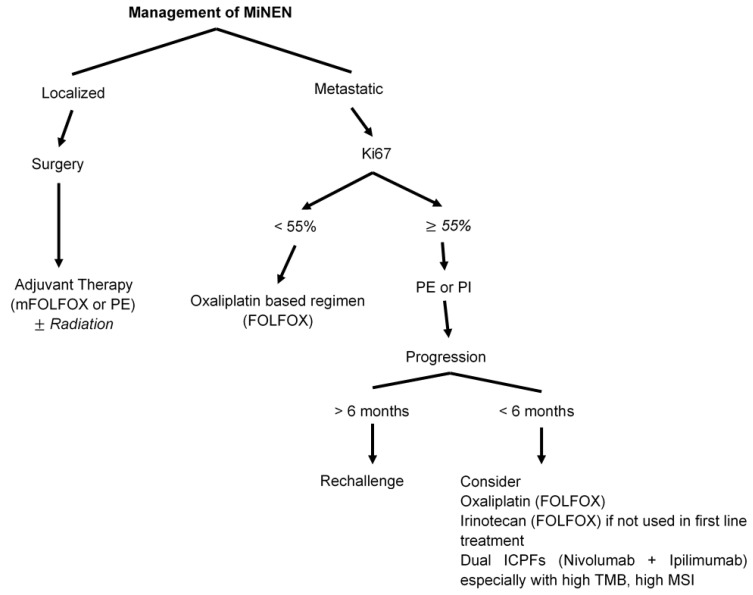
Management of mixed neuroendocrine-non-neuroendocrine neoplasms (MiNENs).

**Table 1 cancers-15-00295-t001:** Appendiceal Neuroendocrine Tumors-specific survival rates [68,74,81].

Stage	TNM	5-Year OS %	10-Year OS %
I	T1N0M0	100	100
II	T1N1M0 T2N0M0	100	100
III	T2N1M0 T3, any N, M0	78	63
IV	Any T, Any N, M1	32	17

**Table 2 cancers-15-00295-t002:** Management Strategies for localized Appendiceal Neuroendocrine Tumor According to Different Guidelines [12,68,74,75,82,83].

Guidelines	Lesion ≤ 1 cm	Lesion 1–2 cm	Lesion ≥ 2 cm
**NCCN**	Appendectomy	Appendectomy	RHC
**NANETs**	Appendectomy	Appendectomy Discuss RHC if tumor at base with HRF	RHC
**ENETS**	Appendectomy Discuss RHC if tumor at base with HRF or R1	Appendectomy Discuss RHC if tumor at tip or middle with HRF Consider RHC if tumor at base with HRF or R1	RHC
**UK NETs**	Appendectomy	Appendectomy Discuss RHC if tumor at base with HRF	RHC

RHC: Right hemicolectomy, HRF: High risk features (>3 mm mesoappendix invasion, positive lymphovascular or perineural invasion, grade 2), NCCN: National Comprehensive Cancer Network, NANETS: North American Neuroendocrine Tumor Society, ENETS: European Neuroendocrine Tumor Society, UK NETs: United Kingdom Neuroendocrine Tumor Society.

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
