# Peer review of "Management of Appendix Neuroendocrine Neoplasms: Insights on the Current Guidelines"

_cancers, 2022, doi:10.3390/cancers15010295_

Round 1

Reviewer 1 Report (Previous Reviewer 3)

Although the authors have not focused on the majortity of appendiceal neoplams (ANET) where considerable debate regarding optimal manegement exist the manuscript covers the whole spectrum of appendiceal neoplasms

ENETS not ENETs

Author Response

  • Although the authors have not focused on the majortity of appendiceal neoplams (ANET) where considerable debate regarding optimal manegement exist the manuscript covers the whole spectrum of appendiceal neoplasms
  • Thank you for the comment. This article only covered different subtypes of appendix neuroendocrine neoplasms and not other appendix neoplasms. We made sure to add all the available data for this rare disease in both localized and metastatic setting.

  • ENETS not ENETs
  • Thank you for the comment. It has been changed to ENETS on pages 8, 9, 10, 21 (Table 2)

Reviewer 2 Report (Previous Reviewer 2)

Please add your remarks on peritoneal metastases to the paper.

Author Response

  • Please add your remarks on peritoneal metastases to the paper.               § 
  • Thank you for the comment. This paragraph has been added to the manuscript on page 12-13 at the end of surgery in metastatic disease: (Other sites of metastases such as peritoneal metastases are likely more common in mucinous appendiceal epithelial neoplasms as compared with ANETs. Depend on data from other appendiceal neoplasms and colon cancer, debulking may improve the overall prognosis and survival of patients with exclusively peritoneal metastases but no current data are available for ANETs . Due to the limited evidence of the overall course and outcome of surgical intervention in this subgroup of patients, there is no agreements or specific guideline established for peritoneal metastasis in the setting of ANETs.) 

Reviewer 3 Report (Previous Reviewer 1)

The authors have insufficiently addressed comments made by the reviewer. Most important, their aim was to report on neuroendocrine tumors of the appendix (ANET), however in their manuscript they refer not only to ANET but also to other tumors of gastro-entero-pancreatic system. 

They extensively report on liver surgery, mTor inhibitors, and other targeted therapies although these therapies are very rarely indicated in ANET. 

The authors should be asked again to revise their manuscript and focus on ANET and their management. 

Author Response

  • The authors have insufficiently addressed comments made by the reviewer. Most important, their aim was to report on neuroendocrine tumors of the appendix (ANET), however in their manuscript they refer not only to ANET but also to other tumors of gastro-entero-pancreatic system.They extensively report on liver surgery, mTor inhibitors, and other targeted therapies although these therapies are very rarely indicated in ANET. The authors should be asked again to revise their manuscript and focus on ANET and their management.
  • Thank you for the comment. I disagree with the reviewer comment because we made sure to answer all the questions in detail and sufficiently (please see our previous answers below). This review article summarized all the current data for managing both localized and metastatic ANENs. The majority of available data are available for GEP-NETs in the metastatic setting with no single prospective trial exclusively for ANENs given the rarity of the disease. That being said, management of metastatic well differentiated ANETs should follow the same algorithm of metastatic well differentiated GEP-NETs include options of LDTs, SSAs, targeted therapies and even PRRT (please refer to NCCN, NANETS and ENETS guidelines). We made sure that for all the current data available for managing metastatic well-differentiated NETs we included how many patients with ANETs in these trials. Please review pages 13-18, which included how many patients with ANETs in these trials if the information was provided in these papers. I will appreciate the reviewer to point if there any other recommendations or prospective trials available exclusively for treating metastatic ANENs.

 Previous comments for reviewer 3 and our answers to every one of them 

1-      This manuscript addresses the management of ANEN based on the WHO 2019 classification but references that include a heterogeneous population of neoplasms are used and thus the data provided do not discriminate between different tumor entities. The authors should focus on NETs (using recently published papers with clearly defined histopathological criteria as Refs 68.69), NEC and MiNEN separately. §  Thank you for the comment. This review article is meant to focus on the histologic subtypes of well-differentiated appendiceal NET, and there is very limited evidence of poorly differentiated NEC or MiNEN involving the appendix. However on WHO for 2019 the terminology well differentiated NETs and poorly differentiated NEC included the whole gastroenteropancreatic group including the appendix.                                           

2-      The authors have provided data for the management of NETs that do not account for appendiceal NETs as defined in the WHO classification that are mostly < 2m, grade 1 (85%), and < 1% grade 3 (Ref 68, 69, Brighi et al, Alexandraki et al). The important issue in this group that is the largest is to identify the most important risks for residual disease or risk metastases necessitating further operation and the follow-up process. The question of lymph node involvement found in these patients and its impact needs to be addressed. In this context the algorithms and recommendations need to be restructured

  • Thank you for your comment. We updated our paragraph in the treatment of localized disease to discuss these risk factors in patients with ANETs less than 2 cm.
  • However, the current recommendations from NANETs, NCCN and other societies guidelines emphasize that surgical management is sufficient for these patients and whether RHC is needed or not still depend on the tumor size (1 cm vs 1-2 cm) which is still considered an area of debate due to lack of prospective trials.

3-      The management of NETs is extensive and not relevant to appendiceal NETs as there were not included in the recent studies whereas metastatic disease is extremely rare in this group of patients.

  • Thank you for the comment. The SEER data collected between 1973-2012 has demonstrated that 12% of Appendix NEN presented with distant metastasis, while no further histologic or grading information are available. We are reporting the management of metastatic disease depend on the current available data, which included most of the patients with GEP-NET. Therefore, we have discussed in every study how many patients with ANENs were included.

4-      In addition, refreences are not entirely correct as they frequently refer to NETs to papers published long before the ewcwnt classification where the term carcinoid included other neoplasms (including Goblet cell tumors). The same accounts when referring to the SEER database that includes a heterogeneous group of neoplasms and therefore comments as to 5-year survival rates are misleading considering that the 5-year survival rate in formally designated NETs is almost 100% (Ref 68,69)

  • Thank you for the comment. We agreed and have added this limitation accordingly. The lower incidence of appendiceal neoplasm renders the clinical experience and histopathologic characteristics in various published data and clinical databases. We recognize that certain types of appendiceal neoplasm with varying classification systems and nomenclature have led to heterogeneity in data regarding epidemiology, clinical courses, and management outcome of these rare neoplasms. Goblet cell carcinoma is known to differ from ANEN for its immunohistochemical characteristics, disease course, and rare association with hormone hypersecretion syndrome. The re-classification and the consensus of staging for goblet cell adenocarcinoma have reinforced the need for further prospective studies to establish optimal management and understand these rare, yet may be aggressive with significant malignant potential, neoplasm in the appendix.

This manuscript is a resubmission of an earlier submission. The following is a list of the peer review reports and author responses from that submission.

Round 1

Reviewer 1 Report

This is a review on management of appendix neuroendocrine neoplasms (ANEN). It should provide insights in current guidelines.

Overall comment

The review covers different types of ANEN, different treatments, and outcome data related to different types of ANEN. For a reader who is not an expert in the field, this harbours a risk to misinterpret the data and the information provided in this review. Since  >90% of ANEN  are well-differentiated grade 1 or grade 2 neuroendocrine tumours (ANET), a significant portion of the data regarding treatment and outcome is irrelevant. This reviewer would strongly suggest focussing on well-differentiated ANET and heavily truncating sections 8.2-8.8.

Minor comments

There are several inaccuracies, e.g. grade 1 and grade 2 ANETS should not be grouped together as “indolent tumors”. While a grade 1 ANET indeed is indolent, an ANET with KI67 of e.g. 18% is not considered as indolent.

The authors should be critical when presenting data; e.g. the SEERT database data have to be taken with caution. 12% of distant metastases may be the case in G3 NET, NEC, and MINEN, respectively, but not in G1 ANET or G2 ANET.

The product of the tumour is irrelevant for surveillance. Would the authors recommend monitoring glucagon-like peptide in a patient who had appendectomy for ANET or ANEN?

The statement ….symptomatic tumours are larger and associated with a higher incidence of obstruction ….requires evidence / referencing.

Correct terminology should be used – 18F FDG PET instead of FDG PET

Somatostatin-receptor based PET is gold standard in the diagnosis and follow -up of NET. It is not only used in metastatic setting or in patients with carcinoid syndrome

Table 1 and 2 require references

The reference list should be truncated. Several references are not of relevance for ANEN (e.g. #130 refers to pancreatic NET, #105 – how many of the liver tumors where metastases of ANEN in this study?)

Reviewer 2 Report

Thank you for the opportunity to review this manuscript. It is well written, comprehensive and provides an excellent overview of the management of this topic.

I have two suggestions only:

-Provide more detail on the sensitivity/specificity of Ga-68/dotatate scans, and alternative diagnoses. We have experienced false positive results in the past.

-Provide more guidance with regard to peritoneal metastases and the management. What are indications for 'debulking'  or 'palliative resection'? Please add references where appropriate.

Thank you for this contribution. 

Reviewer 3 Report

This manuscript addresses the managment of ANEN based on the WHO 2019 clssification but refernces that include a heterogenous population of neoplasms are used and thus the data provided do not discriminate between diffenet tumor entities. The authors should focus on NETs (using recently published papers with clearly defined histopathological criteria as Refs 68.69), NEC and MiNEN separately. 

The authors have provided data for the management of NETs that do not account for appendiceal NETs as defined in the WHO classification that are mostly < 2m, grade 1 (85%), and < 1% grade 3 (Ref 68, 69, Brighi et al, Alexandraki et al). The important issue in this group that is the largest is to identify the most improtant risks for residual disease or risk metastases necessitating further operation and the follow-up process. The question of lymph node involvement found in these patiensta nd its impact needs to be addressed. In this context the algorithms and recommendations need to be restructured

The management of NETs is extensive and not relevant to appendiceal NETs as there were not included in the recent studies whereas metasatic diesease is exteremely rare in this group of patients. In addition, refreences are not entirely correct as they frequently refer to NETs to papers published long before the ewcwnt classification where the term carcinoid included other neoplasms (including Goblet cell tumors). The same accounts when refrering to the SEER database that includes a heterogenous group of neoplasms and therefore comments as to 5-year survival rates are misleading considering that the 5-year survival rate in formally designated NETs is almost 100% (Ref 68,69)

Specific comments